# SSDeN: Framework for Screen-Shooting Resilient Watermarking via Deep Networks in the Frequency Domain

**Rui Bai [1,2,3] , Li Li [1,*], Shanqing Zhang [1] , Jianfeng Lu [1] and Chin-Chen Chang [4]**

1   School of Computer Science and Technology, Hangzhou Dianzi University, Hangzhou 310018, China
2   Key Laboratory of Brain Machine Collaborative Intelligence of Zhejiang Province, Hangzhou 310018, China
3   Hangzhou Dianzi University, Shangyu Institute of Science and Engineering, Shaoxing 312000, China
4   Department of Information Engineering and Computer Science, Feng Chia University,
    Taichung 40724, Taiwan
*   Correspondence: lili2008@hdu.edu.cn

**Abstract:** Mobile devices have been increasingly used to take pictures without leaving a trace. However, the application system can lead to confidential information leaks. A framework for screen-shooting-resilient watermarking via deep networks (SSDeN) in the frequency domain is put forward in this study to solve this problem. The proposed framework can extract the watermark from the leaked photo for copyright protection. SSDeN is an end-to-end process that combines convolutional neural network (CNN) with residual block to embed and extract watermarks in the DCT domain. We simulate some screen-shooting attacks to ensure the networks embed the watermark robustly. Our framework achieves the state-of-the-art performance on existing learning architectures for screen-shooting-resilient watermarking.

**Keywords:** confidential information; screen-shooting; watermark; CNN; DCT; robustly

## 1. Introduction

Digital media techniques in the network information age provide excellent opportunities for information retrieval, access and spreading, while the electronic reading environment presents difficult- or impossible-to-obtain information on paper. These features have remarkably changed and continue to change the way we read. Notably, screen reading is a new trend that has become a necessity for modern people. The use of TV, computer, laptop computer and mobile phone screens and electronic dictionaries in screen reading has changed the way people read from paper to obtain information. At the same time, taking photographs has become a simple and efficient way of information transmission with the prevalence of mobile cameras, thereby increasing the threat to information security. The increased usage of mobile devices to take pictures without leaving a trace has led to many leaks of internal confidential information. A robust watermarking algorithm that extracts watermarks from screen-shooting photographs is necessary to protect confidential information given that images displayed on the screen can embed imperceptible watermarks.

Several screen-shooting-resilient watermarking schemes have been presented in recent years. Robust watermarking techniques are divided into spatial [1] and frequency domains. Compared with that based on the spatial domain, the frequency-domain watermarking method can achieve stronger watermarking robustness without causing serious image distortion. Frequency-domain watermarking techniques, such as DCT [2,3], QDFT [4] and tensor decomposition [4], can help improve imperceptibility and robustness. Fang et al. [2] introduced an intensity-based SIFT framework to extract confidential watermark information from DCT coefficients of the image displayed on the screen. However, SIFT's keypoints are inconsistent and unstable under different screen-shooting attacks. Ahmadi et al. [3] presented a diffusion watermarking scheme that can embed a watermark in the DCT transform domain via learning. Li et al. [4] proposed a watermarking scheme using learned invariant

keypoints, quaternion discrete Fourier transform (QDFT), and tensor decomposition (TD), which hybrid machine learning and traditional watermarking algorithms use to achieve screen-shooting-resilient watermarking.

At present, the applicability of deep learning technology in the field of watermarking algorithm is growing rapidly, because this technology can effectively solve the process of watermark embedding and extraction [3–13]. Tancik et al. [11] proposed a screen-shooting-resilient watermarking technique that jointly trains the scale distortion, blur, noise attack, JPEG compression, and color distortion attacks, while training the deep learning network to resist common printing, remakes, screen photography and other attacks. Wengrowski and Dana [5] put forward an end-to-end watermarking strategy for the screen-shooting process. Datasets [6] were introduced by the framework when the camera and display are at high-perspective angles to guarantee significantly better results than existing schemes. Zhang et al. [7] offered a watermarking scheme with universal deep hiding (UDH) and proved that only adding a perspective transformation layer between the encoder and decoder can resist distortions introduced by the screen-shooting process to a certain extent. Compared with these schemes, a strategy was proposed as the template generation scheme for creating the watermark-embedded template and then embedding the watermark into the red and blue of the carrier image [8]. The trained deep learning network was applied to extract the watermark in the watermark extraction stage. Fang et al. [9] subsequently established a watermarking scheme named "TERA". Similar to [8], the process of watermark embedding depends on a manual design named the superposition-based embedding scheme, and watermark extraction is designed using a deep neural network (DNN). The extraction algorithm is composed of enhancing, attention-guided [10] and regression networks. HiDDeN [13] applied neural networks are used to realize robust watermarking. Specifically, noise distortion layers are added to enhance their framework.

We choose deep learning to construct an image watermarking framework that considers screen-shooting resilient watermarking complexity because of its recent outstanding performance on image watermarking tasks. Ahmadi et al. [3] proposed a robust watermarking technique via deep learning in the transform domain, but the method fails to resist screen-shooting attacks. Tancik et al. [11] put forward a screen-shooting resilient watermarking scheme in the spatial domain with minimal robustness. Spatial-domain-based watermarking is simple to implement, but it has less information embedded and poor robustness. Transform-domain-based watermarking can provide better embedding capacity and robustness under various attacks. We propose a screen-shooting-resilient watermarking algorithm based on [3,11] using deep networks in the DCT domain.

The main contributions of the paper are presented as follows:

- The framework uses a differentiable noise layer for simulating screen shooting attacks between the embedding and extracting layers in the DCT domain to improve security and robustness.
- The moiré phenomenon is a common screen-shooting attack. We trained a convolutional network to add moiré patterns to images automatically.
- We introduce a loss function based on NC and SSIM to balance the robustness and imperceptibility of the algorithm.
- We apply STN before extracting the watermark to correct the image with geometric attacks and enhance the precision of watermark extraction.
- The proposed end-to-end blind robust watermarking framework reduces the artificial pretreatment and post-processing to ensure that the model can change the original input to the final output. Therefore, the model is given additional space to adjust automatically according to image features and improve its overall fitting.

The remainder of this paper is organized as follows. Relevant techniques are described in Section 2. Details of the proposed framework are provided in Section 3. The experimental results are discussed in Section 4. Finally, the conclusions of this study are summarized in Section 5.

## 2. Related Works

We summarize the key issues in designing the SSDeN framework in this section.

### 2.1. Watermarking Algorithm in the Transform Domain

The transform domain watermarking technique can help improve the imperceptibility and robustness of watermarking, and many watermarking schemes, such as DWT [14,15], DFT [16], DCT [17,18], quaternion discrete Fourier transform (QDFT) [19–21] and quaternion Hadamard transform (QHD) [22], have been proposed. Discrete cosine transform (DCT) is a widely used transform domain watermarking technique that modifies the image from the spatial domain to the frequency domain. DCT divides the image into different frequency bands, namely, high-, medium- and low-frequency bands, to select frequency bands easily when embedding the watermark. The middle-frequency band is mainly selected in this work because it can balance the relationship between robustness and imperceptibility and avoid diffusion to high-frequency components against compression and noise attacks to improve image robustness [2]. Watermarking algorithms for different carriers can be divided into two categories in the frequency domain, namely, single- (SCI) and multi-channel images (MCI).

Watermarking-algorithm-based SCI. Guan et al. [23] designed a robust digital image watermarking algorithm that can resist image attacks and distortions and embed the watermark into the two-level DCT coefficients of the image using certain rules. A robust watermarking scheme based on the wavelet transform domain was proposed to resist attacks for single-channel images effectively [24]. A novel image watermarking scheme based on singular value decomposition (SVD) was proposed [25] to overcome the FPP problem and meet the watermarking requirements using integer wavelet transform (IWT). Multi-objective ant colony optimization (MOACO) is utilized to balance robustness and imperceptibility and find the optimal scaling factor while completely avoiding the FPP problem. Rastegar et al. [26] embedded a gray-scale watermark into single-channel images by blending singular value decomposition and the steerable pyramid wavelet transform domain. A robust watermarking method with SVD and DWT [27] was put forward to embed the watermark into the singular values of DWT subbands of the gray image.

Watermarking algorithm-based MCI. Watermarking schemes for color image have continuously emerged because color images are widely used in network transmission and storage [15,19,21,28–35]. Chou and Liu [2] established a new color-image watermarking algorithm based on wavelet transform and significant difference, as well as embedded the maximum watermark information under imperceptible distortion. A link was established between the QDFT coefficient, and the color channel's DFT and QDFT coefficients were modulated to embed the watermark in the transform domain [19]. Wang et al. [21] embedded a watermark into the cover image by adaptively modulating the real coefficients of the quaternion Fourier transform while using LS-SVM to correct the attacked images. Ma et al. [28] developed a local QDFT watermarking method for color images that utilize the advantage of QDFT to improve the robustness and imperceptibility of watermarking. Kais Rouis et al. [29] developed an image hashing algorithm dependent on the analysis of local geometric changes of multi-channel images via gradient measurement for image tempering detection. A geometrically corrected robust color image watermarking method based on the quaternion exponential moments was proposed [30]. Feng et al. [33] combined Tucker decomposition with A-LQIM for a robust image watermarking approach with satisfactory performance. Lorenzo et al. [35] proposed a watermarking method based on the Neyman–Pearson rule for multi-channel images to reduce the error rate when the watermark is nonexistent and to conduct blind image watermarking for gray-scale images in the DCT transform domain.

### 2.2. Watermarking Deep-Learning-Based Algorithm

Several deep learning (DL) watermarking algorithms have been proposed [3,11,13]. HiDDeN [13] uses an algorithm for digital image watermarking that applies a neural net-

work to train an end-to-end model. The algorithmic network allows the adaptive balancing of watermark properties, such as capacity, confidentiality and robustness, to different types of noise by adjusting model parameters or noise layers during training. HiDDeN presents excellent advantages in robust digital image watermarking because new attacks can be directly added to the training process to improve the generalization of the algorithm. The model can robustly hide all kinds of image deformations by considering several different types of image attack layers, unchanged watermarked images, dropouts, crop-outs, cropping, Gaussian and JPEG.

StegaStamp [11] also presented an end-to-end algorithm based on deep learning. StegaStamp learns the watermark information embedding and extracts model resistant image perturbations to simulate the distorted space produced by real printing and photography, including changes in illumination, shading, movement, perspective, rotation, occlusion and viewing distance. Specifically, the researchers simulated five different types of attack layers, including perspective distortion, blurring, color manipulation, Gaussian noise and JPEG, and applied various kinds of image distortions. Finally, robust information extraction performance over multiple printers, screen-shooting and cameras was demonstrated in the experiments.

ReDMark [3] developed a deep end-to-end propagation watermarking algorithm that can flexibly transform the space and learn a new watermarking model. The model consists of CNN with residual blocks to achieve watermark embedding and blind extraction in the simulation environment. The whole network system consists of three main modules: embedding watermark convolution layers, simulating differentiable attack layers and blind extracting watermark convolutional layers. The developed model simulated multiple types of image attacks as a differentiable attack layers to benefit end-to-end training. The watermarking information can be spread in a wide area of the image to enhance the security and robustness of the watermarking algorithm.

StegaStamp used CNN networks that learn a robust encoding and decoding algorithm in the spatial domain and simulate the distortions caused by physically displaying and imaging and image perturbations to train the network for outstanding resistance to screen-shooting attacks. Many researchers proposed the watermarking-based spatial domain and presented weak robustness. ReDMark proposes a deep end-to-end diffusion watermarking framework, which can learn watermarking algorithms in the frequency domain space. However, ReDMark is unable to resist the screen-shooting attacks. The proposed hybrid ReDMark framework and screen-shooting attacks, similar to StegaStamp's distortion [11], is used to train a robust screen-shooting-resilient watermarking.

*2.3. Screen-Shooting Performance Influencing Factors and Capture Device Specifications*

Screen-shooting attacks come from a variety of sources. We analyzed the impact of attacks on real shooting situations. We simulate attack layers on the basis of different cameras, displays and shooting environments using neural networks.

The attributes of display monitors exert a serious impact on the quality of the captured images. Some examples of monitor brands are, "Dell", "Samsung", "LG", "ASUS", "HP", "Huawei", "Acer", "BenQ", "MSI", "Philips", "Lenovo" and "AOC". However, there are general attributes, such as luminosity, the color spectrum of the emitted light, the resolution of the screen and display equipment, that differ for each band.

Phone models also present an impact on the quality of the captured images. Many models, such as "SAMSUNG", "APPLE", "HUAWEI", "NOKIA", "SONY", "LG", "HTC", "XIAOMI" and "LENOVO", are available on the market. Different phone models use various sensors, and corresponding perspective and lens attacks may occur when the image displayed on the screen is captured by the camera equipment and then projected onto a different sensor. Furthermore, the camera device can generate nonlinear lens distortions when the sensor chip is not completely aligned with the imaging and cause radial and tangential lens attacks. The influence of perspective and lens attacks is very important because these attacks can modify the images and watermarks and further influence the

extraction watermark due to the missing critical synchronization information. In addition, barrel attack was caused by the distance of the captured image and the angle of view of the image.

We create a dataset of moiré image pairs using many different combinations of video cameras and displaying devices to fine-tune the model and simulate other screen-shooting attacks in the framework of the watermarking algorithm to address the affect of device differences on screen-shooting resilient watermarking framework.

### 2.4. Multiple Real Leak Cases

Taking photos has become an efficient way of transmitting information with the prevalence of smart phones. Anyone can access files and steal information by taking photos without leaving any records. Several leaked cases, including photographs with the front view, vertical perspective angle and horizontal perspective angle, as well as photos taken from a far distance, are shown in Figure 1.

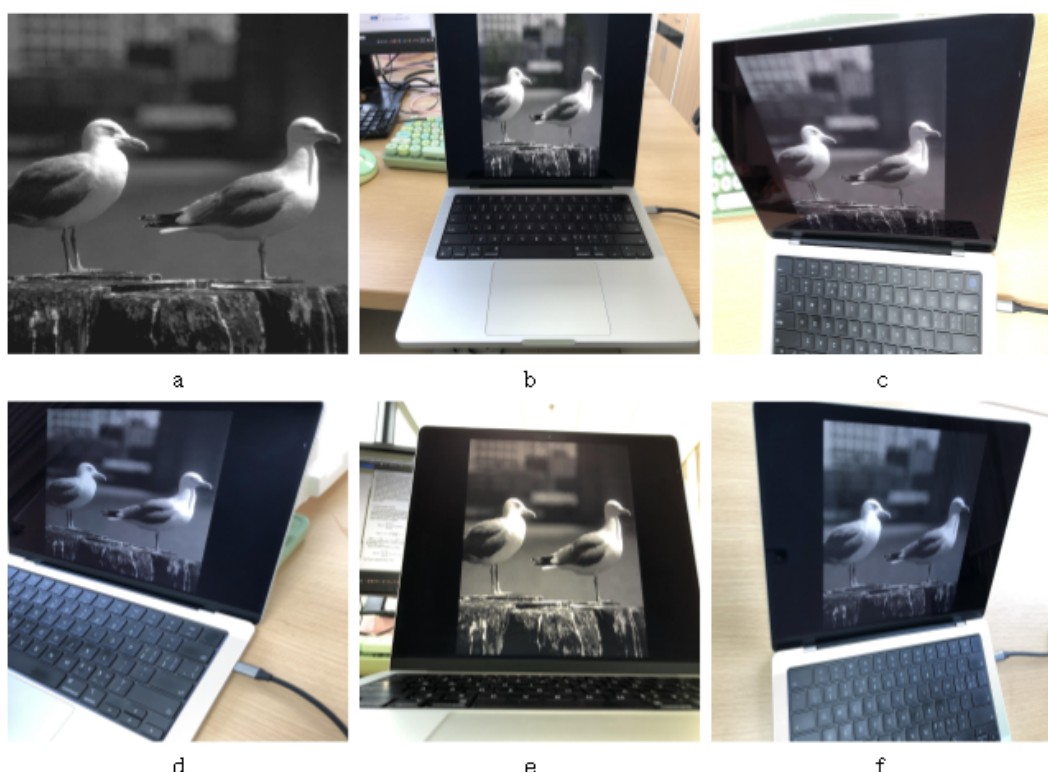

**Figure 1.** Real leak cases: (**a**) original image, (**b**) screen-shooting image with front view, (**c**) screen-shooting image with vertical perspective angle (up), (**d**) screen-shooting image with horizontal perspective angle (right), (**e**) screen-shooting image with vertical perspective angle (down), (**f**) screen-shooting image with horizontal perspective angle (left).

### 3. Method

We discuss screen-shooting-resilient watermarking via deep networks in the DCT domain in this section. The flowchart of the algorithm is shown in Figure 2. The proposed end-to-end framework can achieve blind robust watermarking for gray images in the DCT domain, and simultaneously simulate a differentiable attack layers as part of the framework to implement robust watermarking resistance from various screen-shooting attacks. We will describe the framework and its capabilities in detail.

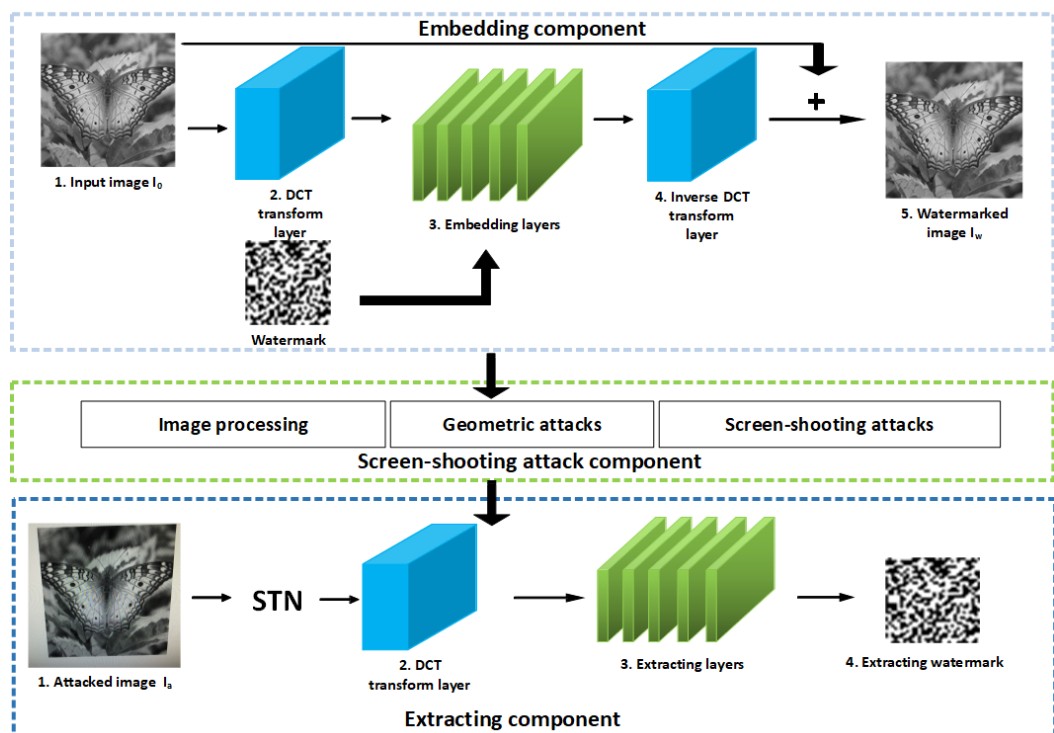

**Figure 2.** The flowchart of the SSDeN framework.

## 3.1. Embedding Component

The embedding component can embed a watermark into the original image to minimize the perceptual difference between the original and the watermark images and improve the imperceptibility and security of the watermarked image. The component is composed of DCT transform layers, convolutional layers and IDCT transform layers, responsible for embedding the watermark into the transform coefficients of the images. We use a CNN architecture that receives a one-channel input of $512 \times 512$ pixels and outputs a gray watermarked image. The input watermark is represented as a $32 \times 32$ bit binary, and transformed image blocks are spliced with the watermark image. Specific procedures of watermark embedding is described in Algorithm 1.

---

**Algorithm 1** Embedding algorithm.

---

**Input:** Image $I_o$ size of $M \times M$, watermark $w_o$ size of $h \times g$, strength factor $\alpha$.
**Output:** Watermarked image $I_w$.
**1:** Divide input image $I_o$ size of $M \times M$ into blocks of size $h \times g$ and each block will be embedding at least one watermark bit. Then we can obtain a tensor of size $b \times h \times g$.
**2:** Input the tensor into the DCT transform layer and output DCT coefficients tensor.
**3:** Concatenate the output of the DCT transform layer with the watermark, shaping the input tensor of size $(b+1) \times h \times w$ as the input of the embedding network.
**4:** Get the output of embedding layer, that is watermarked DCT coefficient map.
**5:** Calculate the residual with watermarked DCT coefficient by the inverse transform layer.
**6:** Add the residual to the input image $I_o$ with a strength factor $\alpha$ weight, finally we can obtain watermarked image $I_w$.

---

## 3.2. Transform Sub-Network

We insert the DCT transform and inverse DCT transform layers in our network framework to transform the image/coefficient and obtain the DCT coefficient feature map/image. $f(x, y)$ is the $M \times M$ pixel matrix, and the specific practice details of DCT transform are presented as follows:

$$D(\mu, \nu) = \sum_{x=0}^{M-1} \sum_{y=0}^{M-1} f(x, y) F(x, y). \tag{1}$$

$$F(x, y) = \alpha_\mu \times \cos(\frac{2 \times y + 1}{2 \times M} \pi \times x). \tag{2}$$

where $\alpha_\mu = \sqrt{\frac{1}{M}}, \mu = 0, \alpha_\mu = \sqrt{\frac{2}{M}}, \mu \neq 0$. $D(\mu, \nu)$ is the discrete cosine transform coefficient corresponding to the pixel matrix $f(x, y)$.

Inverse DCT (IDCT) transform is as follows:

$$f'(x, y) = \sum_{x=0}^{M-1} \sum_{y=0}^{M-1} D(\mu, \nu) F^T(x, y). \tag{3}$$

where $f'(x, y)$ is the pixel matrix of IDCT. The DCT transform layer is embedded in the network to realize the image space transform, and the watermark is embedded in the frequency domain to improve the security, robustness and ability to spread the watermark in a wide area of the image.

### 3.3. Screen-Shooting Attack Component

The screen-shooting operation can be perceived as a cross-media transmitting information procedure. This operation is defined as the process of displaying the watermarked image on the screen and then shooting the screen image with a camera phone. The watermarked image can be subjected to some attacks due to lens distortion, camera properties, display effects, the shooting scene, and desynchronization in this process. The image contains screen-shooting special attacks, three-dimensional distortion and other traditional deformations, including geometric deformation and image processing. Thus, the screen shooting process can be viewed as a collection of various attacks.

#### 3.3.1. Perspective Transform

Screen-shooting photos are susceptible to distortion due to the shooting angle as well as the distance between a camera and a screen. This distortion results in the asynchrony of the watermark information, that is, the watermark's position is shifted. Two images in the same plane are associated by homography to strengthen the robustness of the algorithm resisting perspective transformation attacks. Specifically, we randomly modify the four angular positions within a prescribed range and then solve for the homography that maps the original angle to the new position when sampling the homography.

#### 3.3.2. Optical Distortion

Optical deformations are caused by the optical design of the lens. These deformations occur when special lens elements are used to reduce spherical aberrations and other aberrations. Briefly, optical deformations are the error of lens distortion, and the distortion causes the image to become concave or convex.

#### 3.3.3. Light Distortion

Light distortion is caused by the deformation of light and solar eruptions. Light distortion will manifest itself when the camera takes a picture when a bright light or sunburst shines into the camera because the lighting of the shooting scene environment is difficult to control.

#### 3.3.4. JPEG Distortion

The mobile shooting device will compress the image into a JPEG file during shooting to reduce the storage space that the image occupies. We consider the JPEG attack in the process of training the network to ensure that the compressed JPEG image can still extract the watermark.

### 3.3.5. Moiré Pattern

The moiré mode is rendered when the scene contains repeated details and exceeds the resolution of the sensor. As a result, photos that were taken demonstrate wavy patterns. Specific images are shown in Figure 3.

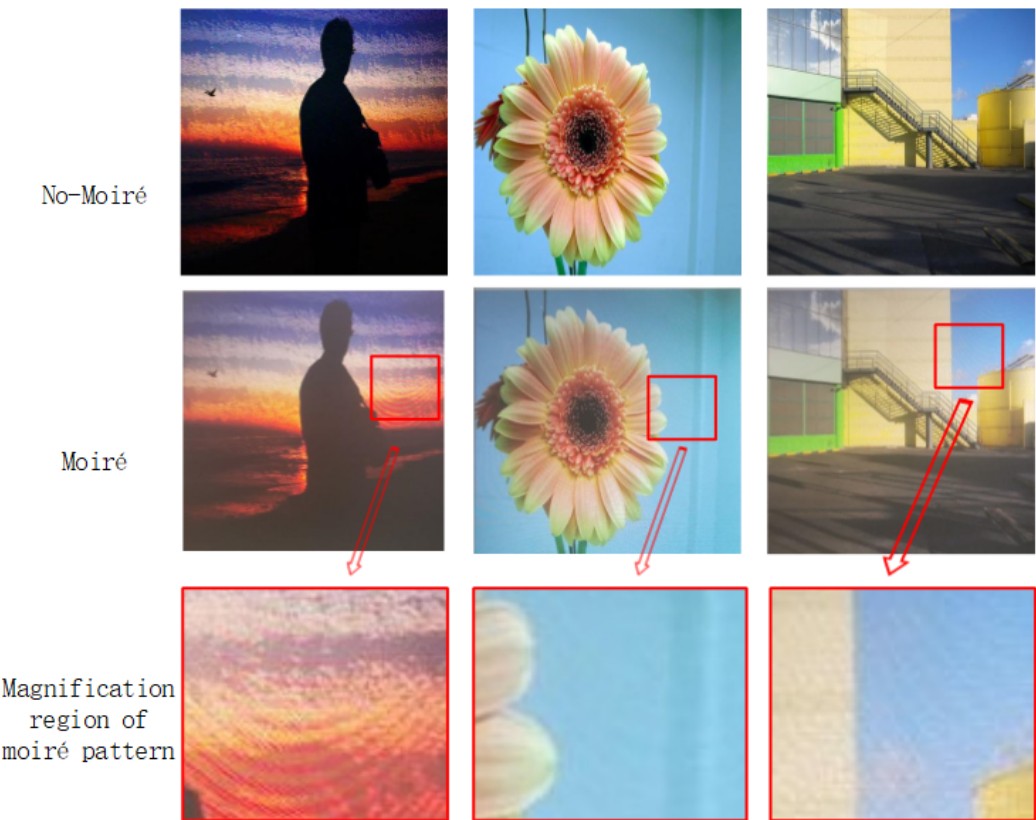

**Figure 3.** (**Row 1**): Pure images. (**Row 2**): Images with moiré pattern. (**Row 3**): A region of the image with moiré pattern and its amplified.

### 3.3.6. Screen Resolution

Different screen resolutions affect the quality of the shot image. Low resolutions will enhance the attack on the shooting image, whereas high resolutions exert a relatively lower impact on the shooting image.

These types of screen shooting attacks were simulated in the network. Although we attempt to reduce other attacks, except device and environment distortions, during the experiment, some effects, such as the special properties of display devices that we ignored in the simulation, were inevitable. In addition, illumination condition, camera positioning and focusing range were different.

In the attack layer, a variety of attacks can be combined to train a robust watermarking network to adapt to the mixture of multiple attacks. The training process is slightly different because in each training iteration, the network randomly selects an attack. Therefore, only one attack is allowed to affect the training loss in each training iteration.

### 3.4. STN

The spatial transformation network (STN) is an adaptive image transformation and alignment network that does not need to calibrate key points. Thus, STN can reduce the influence of geometric transformations, such as rotation, translation, scale, and distortion on the watermark extraction task. Joining an existing network can improve the learning ability of the network.

### 3.5. Extracting Component

We can map the distorted image with the corrected image and then extract the watermark from the corrected image after applying STN. This component extracts the watermark from the watermarked image with screen shooting attacks. The watermarked image is transformed into the DCT domain before the watermark is extracted given that the watermark is embedded in the DCT domain. Since the watermark is embedded in the transform domain, the extraction component contains a copy of the transform layer used to embed the module in order to represent the watermarked image on the same basis. Meanwhile, other network layers learn to extract watermarks in the dct domain. The specific steps of the watermark extraction procedure is described in Algorithm 2.

---

**Algorithm 2** Extracting algorithm.

---

**Input:** Attacked image $I_a$ size of $M \times M$.
**Output:** Watermark $w_e$.
**1:** Apply spatial transformation network to the attacked image $I_a$ size of $M \times M$ and output corrected image.
**2:** Divide corrected image size of $M \times M$ into blocks of size $h \times g$ and obtain a tensor of size $b \times h \times g$.
**3:** Input the tensor into the DCT transform layer and output DCT coefficients tensor.
**4:** Fed the output of DCT transform layer into extracting layers, and then obtain watermark $w_e$.

---

### 3.6. Estimation Criteria

The framework guarantees a watermarked image with maximum quality and minimum watermark extraction error rate to balance the imperceptibility and robustness of the algorithm. Peak signal-to-noise ratio ($PSNR$) [21], structural similarity index ($SSIM$) [3] and normalized cross-correlation ($NC$) [36] are used in this work to evaluate the capability of the SSDeN algorithm. $PSNR$ and $SSIM$ are applied to show the algorithm's imperceptibility performance, and $NC$ is utilized to demonstrate the robustness of watermarking. The mean square error ($MSE$) of the image is expressed as follows:

$$MSE = \frac{1}{M \times M} \sum_{x=0}^{M-1} \sum_{y=0}^{M-1} (I_o(x,y) - I_w(x,y))^2. \tag{4}$$

The $PSNR$ [37] is defined as follows:

$$PSNR = 10 \log_{10} \frac{MAX^2}{MSE}. \tag{5}$$

where $I_o(x,y)$ is the host image, $I_w(x,y)$ is the watermarked image, $MAX$ is the max pixel value of the image:

$$L_I = SSIM(I_o, I_w) = \frac{(2\mu_{I_0}\mu_{I_w} + C_1)(2\sigma_{I_o I_w} + C_2)}{(\mu_{I_o}^2 + \mu_{I_w}^2 + C_1)(\sigma_{I_o}^2 + \sigma_{I_w}^2 + C_2)}. \tag{6}$$

where $I_o$ and $I_w$ denote the original and watermarked images, respectively; $\mu_{I_o}$ and $\mu_{I_w}$ are the average values of $I_o$ and $I_w$, respectively; $\sigma_{I_o}$ and $\sigma_{I_w}$ are the variances of $I_o$ and $I_w$, respectively; $\sigma_{I_o I_w}$ is the covariance of $I_o$ and $I_w$; and $C_1$ and $C_2$ are two variables for stabilizing the division with the weak denominator.

Robustness is used to evaluate the ability of detecting or extracting the watermark from watermarked images after normal image processing or malicious attack. The robustness of the watermarking algorithm is high when the extracted watermark is close to the original watermark. Normalized cross-correlation ($NC$) can be adopted to measure the similarity between the original and the extracted watermarks given that the watermark is a binary image. The value range of $NC$ is [0, 1]. The robustness of image watermarking improves as

the value of $NC$ approaches 1. We choose $NC$ as part of the loss function. $NC$ is defined as follows:

$$L_w = NC = \frac{\sum_{h=1}^{H} \sum_{g=1}^{G} w_e(h, g) \times w_o(h, g)}{\sqrt{(\sum_{h=1}^{H} \sum_{g=1}^{G} (w_e)^2)} \sqrt{(\sum_{h=1}^{H} \sum_{g=1}^{G} (w_o)^2)}} \tag{7}$$

where $w_0(h, g)$ is the original watermark, $w_e(h, g)$ is the extraction watermark and $w_{*(h,g)}$ is a watermark with size $H \times G$.

Finally, the total training loss can be calculated as shown in Equation (8):

$$L_t = \alpha \frac{1}{L_I + C_3} + \beta \frac{1}{L_w + C_4} \tag{8}$$

where $\alpha$ and $\beta$ are the ratios of losses, $\alpha + \beta = 1$ and $C_3$ and $C_4$ are two constants of the metric, which are set to $10^{-4}$ and $9 \times 10^{-4}$, respectively.

## 4. Experiments

The performance of the proposed algorithm is verified using series of experiments, Note that only some representative experimental results are presented. MATLAB R2019b and Python VScode are used in this study as the experimental platforms.

Imperceptibility and robustness are two important evaluation criteria of the watermarking algorithm. As mentioned in Section 3.6, PSNR and SSIM are used to evaluate the quality difference between the original and watermarked images. A large PSNR or SSIM corresponds to enhanced perception quality. The robustness of the algorithm is measured by calculating bit error rate (BER) and NC values. A small BER value or large NC value indicates high robustness. The size of the original image in our experiment is $512 \times 512$, and the size of the watermark image is $32 \times 32$, that is, 1024 bits of binary data.

### 4.1. Simulation Moiré Attack Experiments

The moiré phenomenon often occurs in photographs taken on a computer or television screen with a moving camera. The moiré pattern seriously damages the visual quality of photos and then affects the accuracy of watermark extraction. We train a network for simulating a moiré attack to enhance the robustness of the network. Sun [38] proposed an algorithm that can automatically remove moiré patterns from photographs and obtain pure images using a multi-resolution, fully convolutional network. This objective [38] actually works against our goals. We train a U-Net using a dataset containing 100,000$^+$ images pairs in [38] to simulate the moiré attack layer. Detailed parameters and configurations of the U-Net are listed in Tables 1–3. The training process feeds a pair of images into the moiré pattern attack network, the input is the original image without the moiré pattern attack and the output is the image after the moiré pattern attack. The results of the test experiment are shown in Figures 4 and 5.

Sun [38] built a base library of 135,000 image pairs, with each pair containing an image with the moiré pattern and its corresponding pure image. The image undergoes a series of transformations subjected to a combination of strong attacks, such as color transformations, optical distortion and moiré, when shooting the image displayed on the screen. Hence, the result fails to control a single variable in experiments. Furthermore, the gamut of the monitor is limited compared with the full RGB color space. Cameras use exposure settings, white balance and color correction matrices to modify their output. We demonstrated that the network learns these perturbations and generates images with color manipulation. Some test results are shown in Figure 5.

**Table 1.** Moiré simulating network parameters.

| Epoch | Learning Rate | Layers | Best Loss | Criterion | Batchsize |
|-------|---------------|--------|-----------|-----------|-----------|
| 100   | 0.0001        | 10     | 100       | MSE       | 8         |

**Table 2.** Down-sampling Layers.

| Scale | Kernel | Stride | Channels |
|---|---|---|---|
| 1 | $3 \times 3$ | $1 \times 1$ | 32 |
| 1 | $3 \times 3$ | $1 \times 1$ | 32 |
| 2 | $3 \times 3$ | $1 \times 1$ | 32 |
| 2 | $3 \times 3$ | $1 \times 1$ | 64 |
| 3 | $3 \times 3$ | $1 \times 1$ | 64 |
| 3 | $3 \times 3$ | $1 \times 1$ | 128 |
| 4 | $3 \times 3$ | $1 \times 1$ | 128 |
| 4 | $3 \times 3$ | $1 \times 1$ | 256 |
| 5 | $3 \times 3$ | $1 \times 1$ | 256 |
| 5 | $3 \times 3$ | $1 \times 1$ | 256 |

**Table 3.** Up-sampling Layers.

| Scale | Kernel | | Stride | Channels |
|---|---|---|---|---|
| 1 | $3 \times 3$ | conv | $1 \times 1$ | 3 |
| 2 | $4 \times 4$ | deconv | $2 \times 2$ | 32 |
|   | $3 \times 3$ | conv | $1 \times 1$ | 3 |
| 3 | $4 \times 4$ | deconv | $2 \times 2$ | 64 |
|   | $4 \times 4$ | deconv | $2 \times 2$ | 32 |
|   | $3 \times 3$ | conv | $1 \times 1$ | 3 |
| 4 | $4 \times 4$ | deconv | $2 \times 2$ | 128 |
|   | $4 \times 4$ | deconv | $2 \times 2$ | 64 |
|   | $4 \times 4$ | deconv | $2 \times 2$ | 32 |
|   | $3 \times 3$ | conv | $1 \times 1$ | 3 |
| 5 | $4 \times 4$ | deconv | $2 \times 2$ | 256 |
|   | $4 \times 4$ | deconv | $2 \times 2$ | 128 |
|   | $4 \times 4$ | deconv | $2 \times 2$ | 64 |
|   | $4 \times 4$ | deconv | $2 \times 2$ | 32 |
|   | $3 \times 3$ | conv | $1 \times 1$ | 3 |

Figures 4 and 5 show that the moiré pattern of the standard dataset is weak. Therefore, we build the moiré pattern data set to train the fine-tuned network. We positioned the camera on a tripod to facilitate alignment and steadying. Uncompromised reference images in our benchmark dataset are from the training and test images of the COCO 2014 dataset. We acquired sample pairs with different combinations of phone models and screens to ensure that the capture device shot the moiré pattern on different optical sensors and a diversity of screen resolutions. Phone models were "iPhone 8 Plus", "(MI) Redmi K30" and "HUAWEI nova 8 Pro", and display monitors were "MacBook Pro 14 (2021)", "DELL E2216H" and "AOC E2270SWN5". The training results of the U-Net are presented in Figures 6 and 7. All images used in the experiment were colored, and the proposed watermarking algorithm uses gray-scale images as input images. Therefore, we converted the color image data set of COCO 2014 into gray scale, which was used as the input of the test, and fed it into the moiré network to obtain the texture map with moiré, as shown Figure 8.

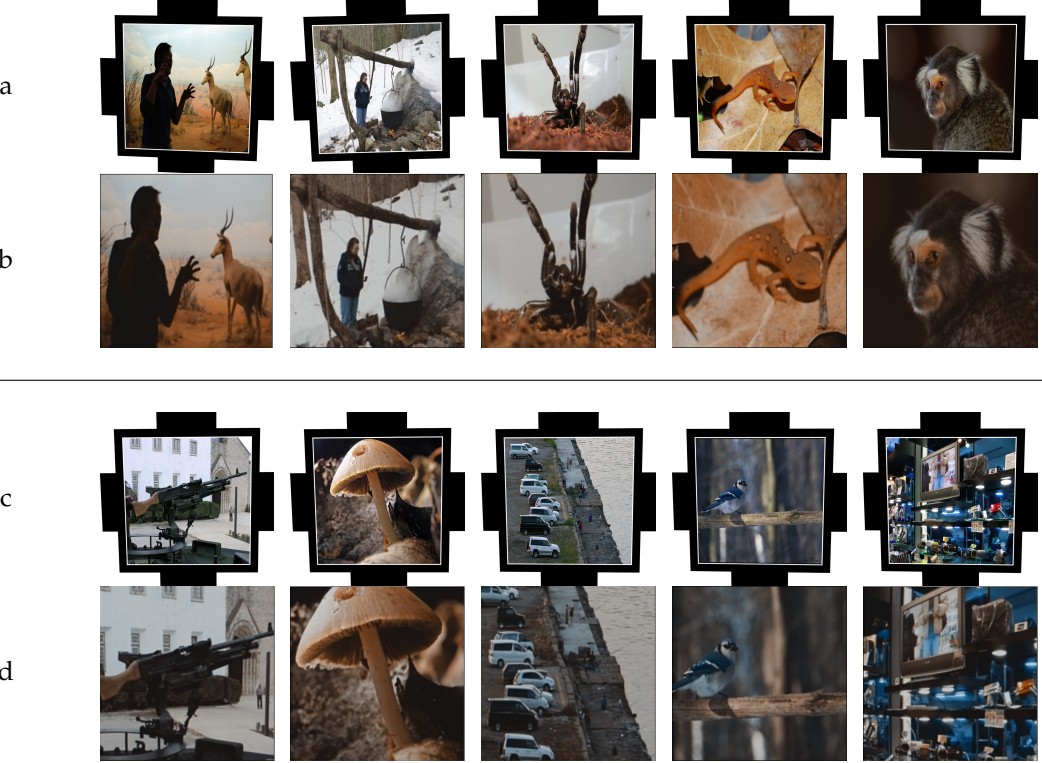

**Figure 4.** Moiré experiment 1. Examples of image pairs from our test experiment: (**a**,**c**) pure images; (**b**,**d**) images (512 × 512) with moiré pattern.

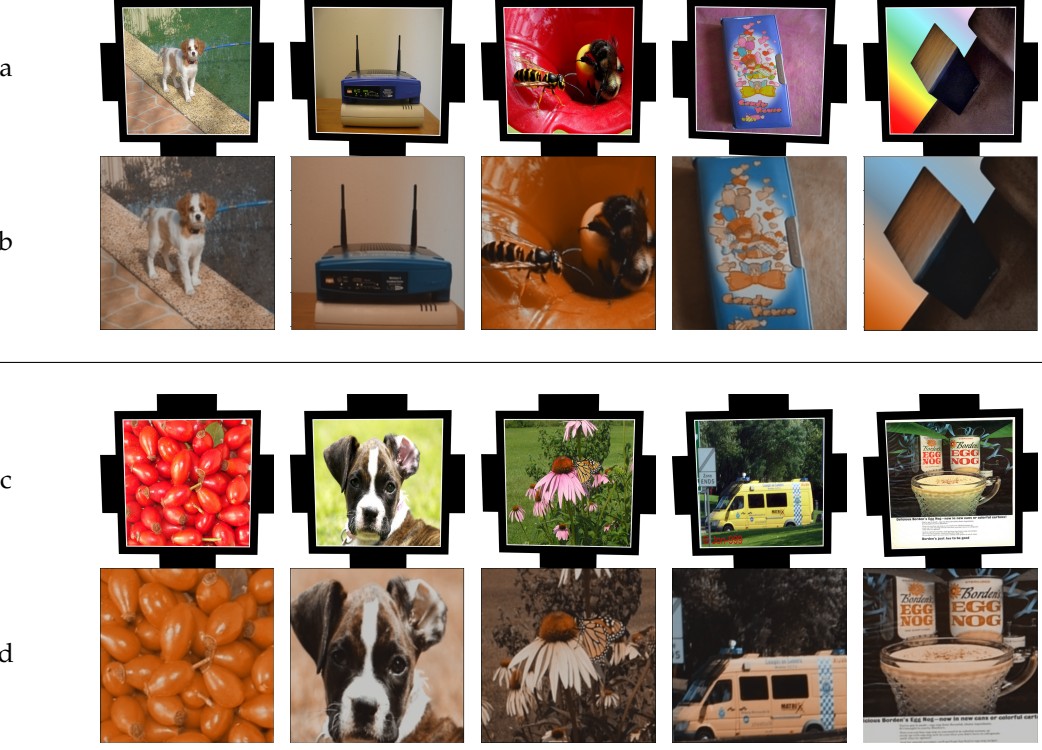

**Figure 5.** Moiré experimental 2. Examples of image pairs from our test experiment: (**a**,**c**) pure images; (**b**,**d**) images (512 × 512) with moiré pattern.

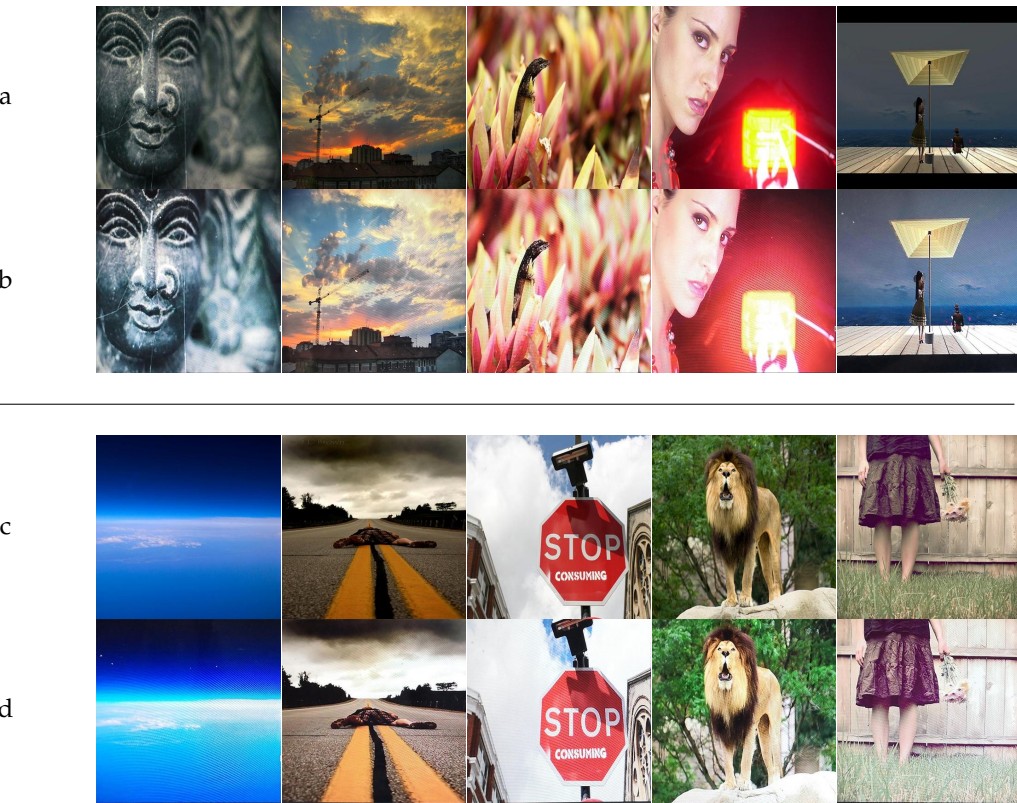

**Figure 6.** Moiré pattern test results from COCO 2014 dataset by the fine-tune network. (**a**,**c**) pure images; (**b**,**d**) images (512 × 512) with moiré pattern.

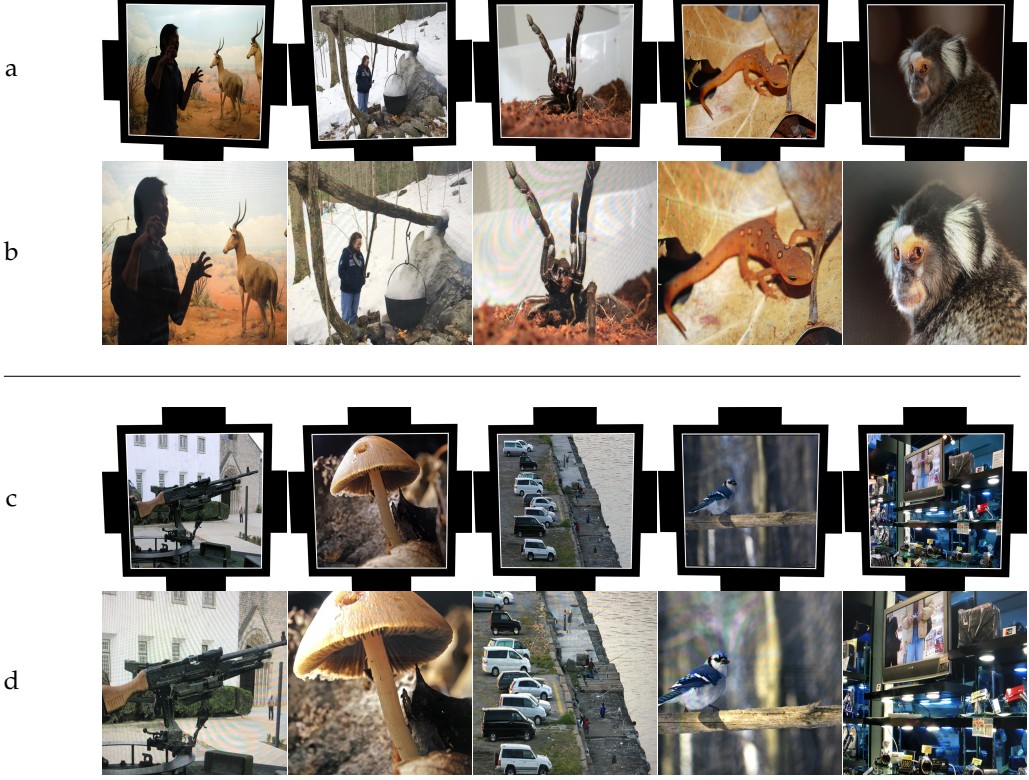

**Figure 7.** Moiré pattern test results from [38] dataset by the fine-tune network. (**a**,**c**) pure images; (**b**,**d**) images (512 × 512) with moiré pattern.

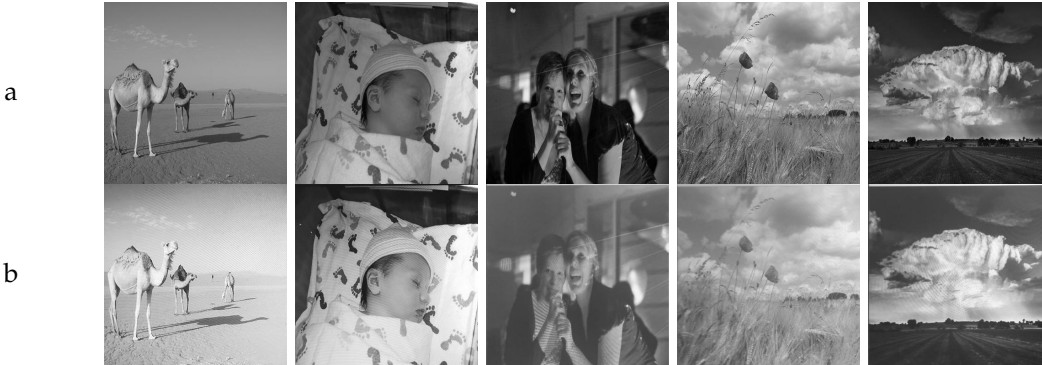

**Figure 8.** Moiré pattern test results from COCO 2014 gray dataset by the fine-tune network. (**a**) pure images; (**b**) images (512 × 512) with moiré pattern.

The moiré network training criterion $L_m$ [38] is as follows:

$$L_m = \frac{1}{m}\sum_{i=1}^{m}(y_i - f(x_i))^2 \tag{9}$$

where $m$ is the number of samples, $y_i$ is the $i$th ground truth, $f(x_i)$ is the $i$th predicted value and $L_m$ is the mean square deviation between the estimated and actual values. It is a positive value and decreases as the error approaches zero because the value is calculated from the square of the Euclidean distance. The detailed training loss values of the training and validation dataset of the moiré network are presented in Figures 9 and 10, respectively. We also measure the PSNR of the test image pair experiments, as shown in Figure 11. PSNR fails to measure the effectiveness in the moiré pattern precisely, and an image corrupted by visually severe moiré patterns can demonstrate high PSNR. However, PSNR can measure the overall image quality and the moiré experimental results remain unaffected.

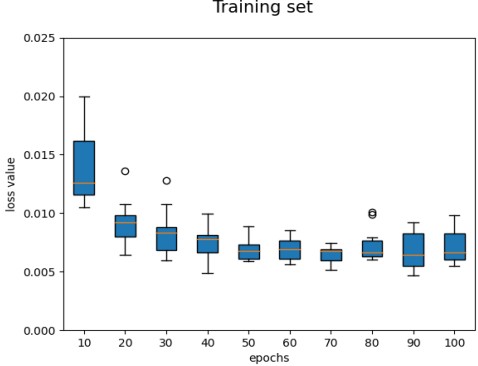

**Figure 9.** Training loss of moiré network.

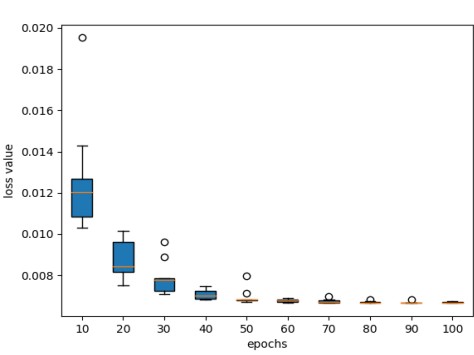

**Figure 10.** Validation loss of moiré network.

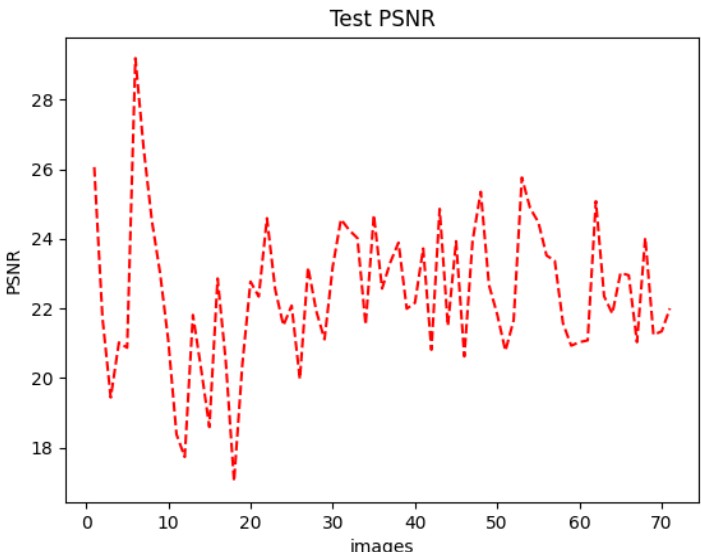

**Figure 11.** PNSR of the moiré network.

We insert the moiré attack layer between the embedding and extracting processes to improve the robustness of screen-shooting resilient watermarking and then add a variety of attack layers.

### 4.2. Optical Effects of Various Simulation Attacks

We show the results of simulation attacks in this part. Various attacks simulated for the attack layer are briefly explained. These distortions are generally classified into two categories: normal and special attacks.

Normal attacks (NA). (A). JPEG (parameters: 30, 50 and 60). Photographs are usually saved in a JPEG compression after being shot, and JPEG compression will cause image loss to the captured photographs. (B). Noise layer. (Gaussian noise: 1, 2, 3, . . ., 10). Image noise represented as random blobs on a smooth surface can severely affect image quality. The intensity of the noise depends on exposure time, camera settings and the temperature. (C). Cropping attack (parameters: 0.06, 0.11, 0.16, 0.26 and 0.55). Only some images on the display will be captured as full images due to the shooting environment. (D). Blurring (parameters: 1, 1.2, 1.6, 2 and 2.8). The inconsistent bending of light leads to blurred images, and motion blur and defocus blurs are the common blur types of blur that degrade image quality. (E). Rotation (parameters: −5, −2, 5, 8 and 10), scaling (parameters: 0.25, 0.5, 1.25, 1.5 and 2) and translation (RST) distortions. RST distortions are common traditional geometric attacks. (F). Brightness and contrast (brightness = 0.3, contrast = 1.5). The whole range of tones in the image will rise or fall accordingly when the brightness is corrected. Midtones are eliminated and images will present a high percentage of dark or black and white or highlights with minimal midtones when the contrast is high. (G). Sharpening (1, 4, 12, 25 and 30). Sharpening is a distortion that increases the apparent sharpness of an image.

Special attacks (SA). The image suffers from space conversion processes and lead to a combination of strong attacks, such as perspective deformation, moiré (U-Net), lens distortion, visible watermark (parameters: 1, 3, 9, 12, 15 and 16), light (brightness and contrast), color changing (hue = 0.1, luminance = 0.5) and display distortion (scaling), when an image displayed on the screen is shot.

We provide attacked images, the specific results of the attack are illustrated in Figure 12.

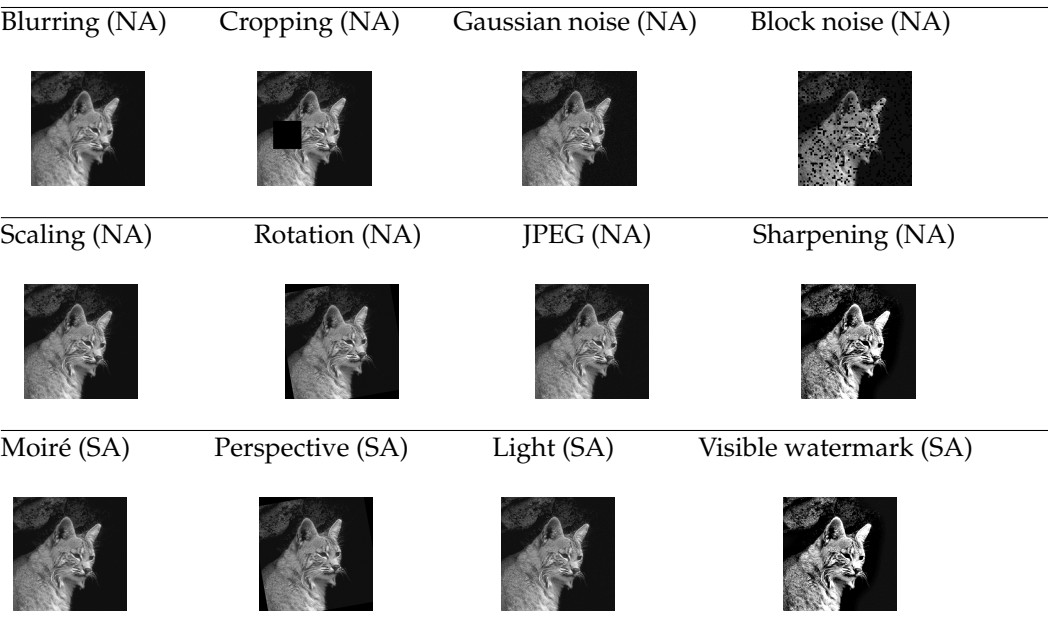

**Figure 12.** Simulating image distortions for screen-shooting.

*4.3. Quantitative Analysis*

We choose 25 images (512 × 512) from the Granada dataset for our test experiment. The fidelity of each watermarked image is evaluated from PSNR and SSIM criteria. Figure 13 shows the PSNR of watermarked images generated by all trained networks with fixed strength factors ($\alpha$ = 1). The average of PSNR is 36.094852.

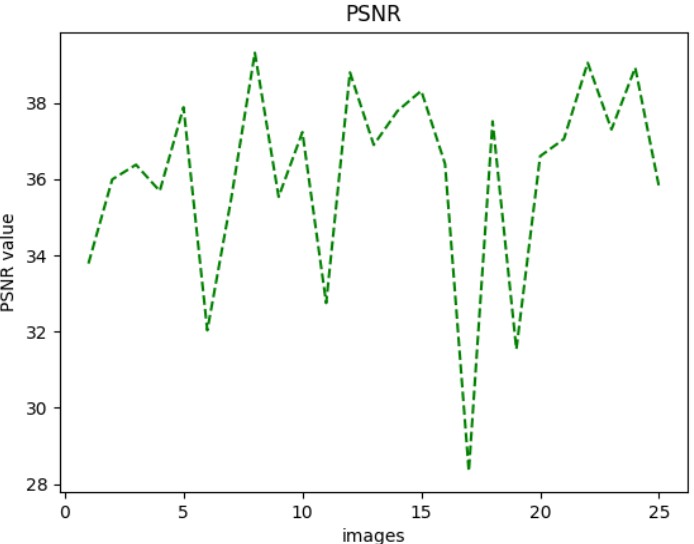

**Figure 13.** Test images (25) PSNR of SSDeN.

The robustness of a watermarking algorithm can be determined using two approaches. One approach is to enhance the watermark embedding strength. High-intensity embedding increases the robustness but reduces the PSNR. We balance the robustness and impercepti-bility of the trained network by adjusting the watermark strength factor ($\alpha$ = 1). The bit error rate of the watermark is calculated under a variety of attacks with fixed strength factors to prove the robustness of the proposed framework. The other approach is to increase the diversity of image attack types to enhance the robustness of the framework, as shown in Table 4. Our framework can effectively embed the watermark and avoid

various image distortions. Figure 14 illustrates the original and watermarked images as well as the difference image between the original image and the watermarked images with an enhancement of 50×.

**Table 4.** For each attack, the attack parameter and the resulting extraction BER are shown for SSDeN.

| Images | Cat | Barbara | Baboon |
|---|---|---|---|
| Attacks | | | |
| **Blurring (1.20)** | 0.015 | 0.020 | 0.016 |
| **Cropping (0.06)** | 0.026 | 0.020 | 0.021 |
| **Gaussian noise (8.00)** | 0.000 | 0.002 | 0.001 |
| **Block noise (0.16)** | 0.011 | 0.019 | 0.000 |
| **Scaling (2.00)** | 0.0001 | 0.004 | 0.001 |
| **Rotation (10.00)** | 0.0483 | 0.0513 | 0.0480 |
| **JPEG (60.00)** | 0.020 | 0.020 | 0.000 |
| **Sharpening (4.00)** | 0.000 | 0.009 | 0.006 |
| **Moireé (U-Net)** | 0.0179 | 0.0201 | 0.0103 |
| **Perspective ($\theta = 5$)** | 0.036 | 0.030 | 0.0298 |
| **Light (0.3, 1.5)** | 0.0015 | 0.0026 | 0.0010 |
| **Visible watermark (16.00)** | 0.010 | 0.0091 | 0.0099 |

Original images

Watermarked images

Residual ×50

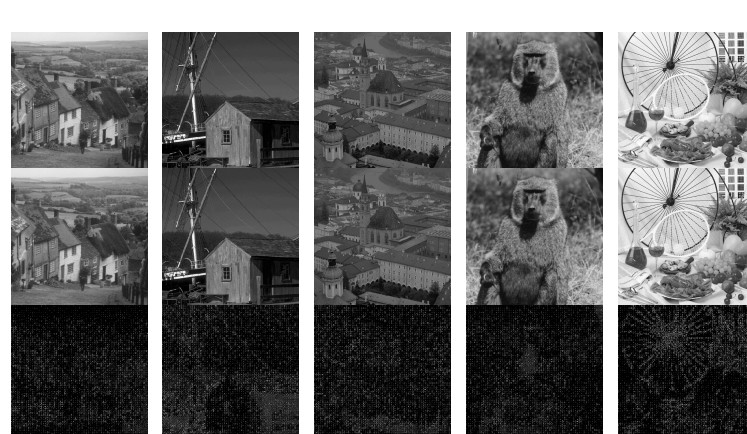

**Figure 14.** The results of images watermark embedding and the residual for original and watermarked images.

### 4.4. Qualitative Analysis

Watermarked DCT coefficients produced by the framework are illustrated in Figure 14. A random 1024-bit (32 × 32) watermark is embedded in the DCT domain of the 'Cat', 'Barbara' and 'Baboon' images from the Granada dataset using the proposed framework. The absolute difference between the original and the watermarked images is used to display the watermark distribution. This difference is multiplied by 10, 20, 50, 80 and 100 to improve the visualization. In addition, a small area of the image is enlarged to facilitate the presentation. We revealed that the amplitude of the resulting artifact varies in different regions of the original DCT coefficients in diverse images. The variation in the difference matrix indicates that the watermark symbol is adaptively embedded on the basis of local features of the image.

Analysis of the diffusion pattern of the DCT domain and its frequency energy map demonstrated that SSDeN adopts a distributed embedding policy concentrated in the middle- and low-frequency bands. As demonstrated in Figures 15 and 16, Figure 16 presents the 3D plot of frequency energy. The frequency energy is calculated using the absolute values of DCT coefficients of an $8 \times 8$ block of the diffusion watermark. DCT coefficients are arranged on the horizontal axis in zigzag order of the DCT block. For example, we start from the DC coefficient DCT(0, 0) toward the maximum frequency DCT(7, 7). The framework learns to avoid embedding in the DC coefficient due to destructive effects on image quality and PSNR.

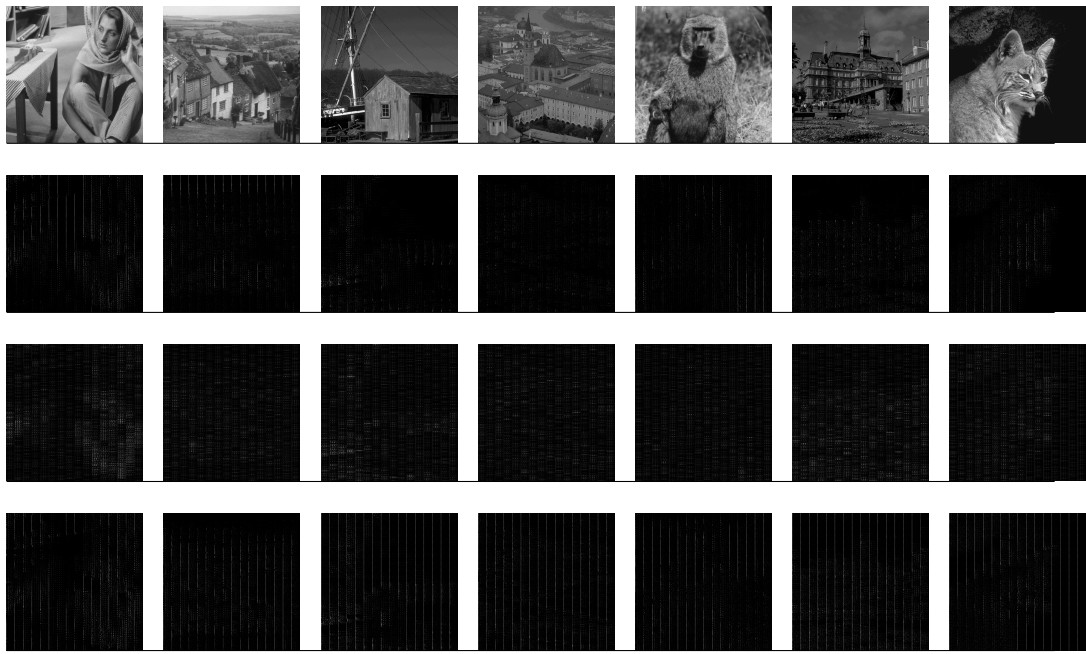

**Figure 15.** The results of the DCT coefficient: (**Row 1**) original images; (**Row 2**) DCT coefficient of original images; (**Row 3**) DCT coefficient of watermarked images; (**Row 4**) the residual for original and watermarked DCT coefficients.

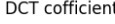
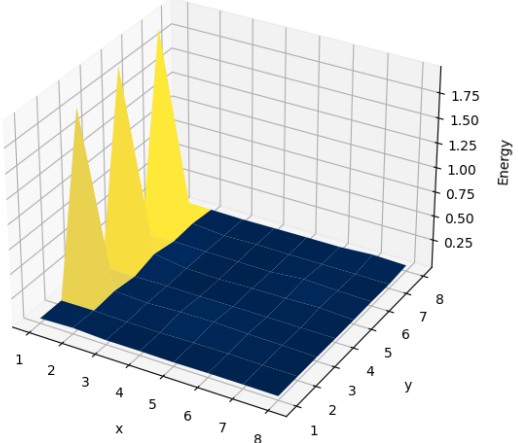

**Figure 16.** DCT energy 3D plot of the framework.

According to the Table 4, the proposed framework achieves an acceptable trade-off between imperceptibility and robustness. The experimental results showed that the proposed algorithm is superior to existing advanced schemes.

## 4.5. Comparison with State-of-the-Art

We exhibit and discuss the comparative experimental results and compare them with other schemes in this section [2,3,11–13,39]. The comparison of watermarked image quality between our scheme and existing schemes, including PSNR, network architecture, watermark embedding domain and published time, is presented in Table 5.

**Table 5.** Methods comparison.

| Methods | Network | PSNR | Published Time | Embedding Domain |
|---------|---------|------|----------------|------------------|
| StegaStamp [11] | U-Net | 29.88 | 2020 | Spatial domain |
| HiDDeN [13] | GAN | 33.55 | 2018 | Spatial domain |
| ReDMark [3] | CNN | 35.93 | 2019 | Frequency domain |
| TSDL [12] | GAN | 33.50 | 2019 | Spatial domain |
| Fang et al. [2] | ISIFT and DCT | 42.30 | 2019 | Frequency domain |
| Nakamura et al. [39] | Geometric distortion | 39.10 | 2004 | Spatial domain |
| The proposed (SSDeN) | CNN | 36.09 | 2022 | Frequency domain |

As Table 5 shows, in recent years, many screen-shooting watermarking methods have been presented [2,3,11–13,39]. Fang et al. [2] and Nakamura et al. [39] were using traditional methods for resistance to screen-shooting attacks; they have good image quality, but the registration point is unstable and may not be able to extract the watermark. As deep learning develops, deep-learning-based watermarking algorithms have been widely studied in various network structures [3,11–13], including U-Net, GAN, and CNN. For robust watermarking, HiDDeN [3] is, to our knowledge, the first end-to-end method using neural networks in the spatial domain, but it could not resist the screen-shooting attack. StegaStamp [11] was the first to propose a machine learning screen-shooting attack algorithm in the spatial domain. The algorithm has good robustness, but the image quality is slightly worse. ReDMark [3] was the first robust watermarking using deep learning in the frequency domain, and it has relatively good image quality; it also could not resist the screen-shooting attack.

Table 5 displays that there have been few frequency-domain watermarking algorithms based on machine learning proposed in recent years, except ReDMark [3]. Watermarking algorithms in the frequency domain can help improve the imperceptibility of watermarking. Frequency-domain watermarking algorithms can guarantee enhanced imperceptibility and robustness of embedding after various attacks. Therefore, we investigate watermarking in the frequency domain via deep learning to improve the performance and efficiency in balancing the relationship between robustness and imperceptibility. However, the framework is limited by its unsuitability for color images.

We also designed experiments to test the robustness of our networks and other existing schemes. The results are listed Table 6; (*) is the parameters used for each type of attack.

The performance of the proposed method is illustrated by comparing with other deep-learning-based watermarking methods [3,11–13], and the advantages of the proposed method are illustrated by experimental results.

StegaStamp [11] proposed an end-to-end watermarking scheme for the print-camera process in the spatial domain. This method demonstrated very high robustness to print-camera distortions. Table 6 shows that StegaStamp has a good performance against perspective and light. However, the method has a poor performance against cropping and JPEG. The overall robust performance is considered to be positive in the spatial domain.The watermark information in ReDMark [3] is spread in a wide area of the image. ReDMark performs well against cropping, scaling, noise and visible watermarking. Compared with

the watermarking method based on the spatial domain, the watermarking method based on the transform domain is more robust. Although this method can resist traditional attacks robustly, it fails to resist screen-shooting attacks. We add the screen-shooting attack layer to train our model and improve the robustness. The results showed that our model presents strong resistance to screen-shooting attacks and traditional attacks. HiDDeN [13] fed augmentations into their training process to increase the robustness against several traditional attacks. The researchers also proposed the watermarking-based spatial domain and, compared with frequency-domain-based methods, presented weak robustness. From various attack results, this method has no outstanding advantage. However, HiDDeN [13] was the first end-to-end robust method using neural networks and provided lessons on deep-learning-based watermarking. The two-stage separable deep learning (TSDL) [12] scheme for robust watermarking is composed of noise-free end-to-end adversary training and noise-aware decoder-only training and robust to several black-box noises. TSDL excels in blurring and color changing for color image. The spatial-domain-based watermarking usually perform poorly in JPEG and cropping, so the same goes for TSDL, and it is not resistant to screen-shooting attacks.

**Table 6.** Comparing the extracted watermark of the proposed algorithm and previous methods in terms of NC.

| Methods | StegaStamp [11] | HiDDeN [13] | ReDMark [3] | TSDL [12] | The Proposed (SS-DeN) |
|---|---|---|---|---|---|
| Cropping (10) | 0.7697 | 0.8823 | **0.9815** | 0.8501 | 0.9710 |
| JPEG (50) | 0.7952 | 0.7731 | 0.9779 | 0.7770 | **0.9798** |
| Blurring (1.2) | 0.9142 | 0.7553 | 0.7871 | **0.9956** | 0.9850 |
| Scaling (2.0) | 0.9987 | NULL | **0.9995** | 0.9233 | 0.9800 |
| Sharpening (4) | NULL | NULL | 0.9981 | 0.9900 | **1.0** |
| Perspective | **0.9897** | NULL | NULL | NULL | 0.9640 |
| Color changing | 0.9878 | NULL | NULL | **0.9886** | NULL |
| Noise (10) | 0.9523 | 0.9212 | **0.9889** | 0.8995 | **1.0** |
| Moiré (U-Net) | 0.8952 | NULL | NULL | NULL | **0.9926** |
| Visible watermark (15) | 0.9700 | NULL | **0.9980** | 0.9567 | 0.9975 |
| Light | **0.9994** | NULL | NULL | 0.9929 | 0.9899 |

Hence, these experiments demonstrated that the proposed SSDeN framework is robust not only to common traditional attacks but also to screen-shooting attacks.

## 5. Conclusions

We propose an end-to-end, screen-shooting-resilient image watermarking framework using convolution neural network in this study. Our framework consists of four components, namely, embedding, transform sub-network, simulating attack layers and extracting components. The embedding component is used to embed the watermark into the DCT coefficients of images. The transform sub-network modifies the image to obtain the DCT coefficient feature of the image. The simulating attack layer is designed to simulate a series of distortions, such as perspective transform, optical distortion, light distortion, JPEG distortion and moiré pattern, in the screen-shooting process. We design a moiré attack network for simulating the moiré phenomenon, which is the most common screen-shooting attack, to improve the resiliency of the watermarked image against distortions in real screen-shooting scenes. The extracting component extracts the watermark from the captured photos. In order to balance the robustness and imperceptibility of the algorithm, we introduce a loss function based on NC and SSIM. From the experiments, it can be concluded that SSDeN employs a distributed embedding strategy with greater concentration on low- and middle-frequency coefficients in keeping with the traditional method's results. Simultaneously, the above shows the rationality of our proposed framework.

Our follow-up investigation will continue to improve and extend SSDeN. Specifically, we will simulate additional capture devices and increasingly complex shooting scenes to improve watermark detection in real-world scenarios. Screen-shooting-resilient watermarking can also be extended for color images while fully fitting all real screen-shooting scenarios.

**Author Contributions:** Conceptualization, S.Z. and R.B.; methodology, L.L.; software, R.B.; validation, R.B., L.L. and S.Z.; formal analysis, C.-C.C.; investigation, R.B.; resources, R.B.; data curation, R.B.; writing—original draft preparation, C.-C.C.; writing—review and editing, L.L.; visualization, J.L.; supervision, R.B.; project administration, R.B.; funding acquisition, L.L. All authors have read and agreed to the published version of the manuscript.

**Funding:** This research was funded by the National Natural Science Foundation of China (No. 62172132) and the Public Welfare Technology Research Project of Zhejiang Province (No. LGF21F020014).

**Institutional Review Board Statement:** Not applicable.

**Informed Consent Statement:** Not applicable.

**Data Availability Statement:** Not applicable.

**Acknowledgments:** Thank you to the reviewers who reviewed this paper and the MDPI editor who edited it professionally.

**Conflicts of Interest:** The authors declare no conflict of interest.

**Sample Availability:** Not applicable.

## Abbreviations

The following abbreviations are used in this manuscript:

| | |
|---|---|
| CNN | Convolutional neural network |
| DCT | Discrete cosine transform |
| IDCT | Inverse discrete cosine transform |
| QDFT | Quaternion discrete Fourier transform |
| FRFS | Feature regions filtering model to SuperPoint |
| CDTF | Camera-display transfer function |
| UDH | Universal deep hiding |
| JPEG | Joint photographic experts group |
| DNN | Deep neural network |
| TERA | Transparency, efficiency, robustness and adaptability |
| NC | Normalized correlation |
| SSIM | Structural similarity index |
| DWT | Discrete wavelet transform |
| DFT | Discrete Fourier transformation |
| SCI | Single-channel images |
| MCI | Multi-channel images |
| SVD | Singular value decomposition |
| FRAT | Finite radon transform |
| LS-SVM | Least squares support vector machines |
| QR | Quadrature rectangle decomposition |
| DL | Deep learning |
| PSNR | Peak signal-to-noise ratio |
| MSE | Mean square error |
| NA | Normal attacks |
| RST | Rotation, scaling and translation |
| SA | Special attacks |
| BER | Bit error rate |
| TSDL | Two-stage separable deep learning |
| TD | Tensor decomposition |
| SIFT | Scale invariant feature transform |

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
