# Peer review of "SSDeN: Framework for Screen-Shooting Resilient Watermarking via Deep Networks in the Frequency Domain"

_applsci, doi:10.3390/app12199780_

Round 1
Reviewer 1 Report
This paper is about a screen-shooting resilient watermarking scheme. The authors propose using deep neural networks and frequency domain to achieve robustness in the D/A and A/D conversion. The subject is very interesting by itself, and practical applications arise quickly. However, the manuscript seems immature; some of my concerns are as follows:
- Figures 1-3 are very ambiguous; it is unclear what those mean.
- Authors lack a deep explication of their scheme. Figure 5 is very poorly given. It is necessary to detail with a pseudo-code the whole process, from training to deployment. Flow charts are also desirable among the previous pseudo-code.
- It is unclear which task the NN does and which does not. Which is the embedding rule? How is the embedding rule related to the NN?
- There are a lot of typos within the manuscript; some text from the template must be edited.
Author Response
I have uploaded the revised manuscript, thank you.

Reviewer 2 Report
The stages of the proposed method should be explained in more detail. How is it different from the previous method? What solutions are given to your proposed method to overcome the problems in the previous methods?
You need to adjust the order of the explanations in Section 3 with Fig. 5. Explain in more detail the number of layers used, layer order, hyperparameters, and so on. Of course, you don't just use standard DCT for embedding or STN for extraction to produce a reliable method
Based on Fig. 5, it can be seen that the watermark used is a watermark inputted by the user, not generated from a certain method. Then why choose an abstract watermark? Because visually, it is difficult to see the difference.
It is not correct if Eq. (5) using 255, a more precise value is max, because the max pixel value of the image used is not necessarily 255, see 10.1007/s11042-020-10035-z
You describe the SSIM measurement tool, but no SSIM measurement data is presented
If you use Granada dataset, why is it presented instead from fine-tune network and COCO 2014? Were the fine-tune network and COCO 2014 datasets also used in this research? It is also necessary to present a sample dataset used in addition to the Cat image. In addition, it is better to provide a link to the dataset if it is publicly accessible.
How do you do the comparison? Need to be explained in more detail. In particular, the data is presented in Table 4 and Table 5.
Fig. 17 and 18 also need to be presented with NC values ​​, and the parameters used for each type of attack also need to be explained. Fig. 17 & 18 is also more appropriate if made into a table.
The attack example presented in Fig. 18 is quite different when compared to Fig. 6. Although the goal may be later so that a comparative test can be carried out. But real testing should also be used.
The types of attacks presented in Table 5 need to be explained further with what parameters are used. For example, what noise is used, what size scaling is used, etc
Author Response

(The authors gave the same response as above.)

Round 2
Reviewer 1 Report
The authors made a good effort to improve the manuscript. I suggest verifying the journal format before being accepted, e.g., correspondence on the front page is incomplete; patents at the end must be modified; etc.
Author Response
Thank you for your letter. I am very pleased to learn that our manuscript is acceptable for publication with revision. And then I will verify and modify the format, correspondence, patents, etc. We should like to thank the reviewer for your helpful comments again.

Reviewer 2 Report
The paper is acceptable, only needs a little writing improvement
Author Response
Thanks for accepting manuscript. I greatly appreciate both your help and that of the referees concerning improvement to this paper. I, meanwhile, will improve the writing.
